# Fact Embedding through Diffusion Model for Knowledge Graph Completion

## ABSTRACT

Knowledge graph embedding (KGE) is an efficient and scalable method for knowledge graph completion tasks. Existing KGE models typically map entities and relations into a unified continuous vector space and define a score function to capture the connectivity patterns among the elements (entities and relations) of facts. The score on a fact measures its plausibility in a knowledge graph (KG). However, since the connectivity patterns are very complex in a real knowledge graph, it is difficult to define an explicit and efficient score function to capture them, which also limits their performance. This paper argues that plausible facts in a knowledge graph come from a distribution in the low-dimensional fact space. Inspired by this insight, this paper proposes a novel framework called Fact Embedding through Diffusion Model (FDM) to address the knowledge graph completion task. Instead of defining a score function to measure the plausibility of facts in a knowledge graph, this framework directly learns the distribution of plausible facts from the known knowledge graph and casts the entity prediction task into the conditional fact generation task. Specifically, we concatenate the elements embedding in a fact as a whole and take it as input. Then, we introduce a Conditional Fact Denoiser to learn the reverse denoising diffusion process and generate the target fact embedding from noised data. Extensive experiments demonstrate that FDM significantly outperforms existing state-of-the-art methods in three benchmark datasets. Especially on FB15k-237, FDM achieves a 16.8% relative improvement in MRR scores compared to the state-of-the-art methods.

## 1 INTRODUCTION

Knowledge graphs (KGs) are a type of multi-relational graph that stores factual knowledge from both the web and the real world. Due to their high efficiency in storing and representing factual knowledge, KGs are essential for many applications such as question answering [14], information retrieval [41], recommender systems [46], and natural language processing [44]. Knowledge graphs are typically stored using the W3C standard RDF (Resource Description Framework) [3], which models knowledge graphs as enormous facts (triplets). Each fact $(h, r, t)$ consists of a head entity $h$, a relation $r$, and a tail entity $t$, representing resources described on the Web. However, due to the complexity of the resources on the Web, knowledge graphs are often incomplete, which restricts their applications in downstream tasks. Therefore, knowledge graph completion (KGC) has been proposed to complete missing facts by inferring from existing ones. Generally, the KGC task is to make the entity prediction for incomplete facts i.e., $(h, r, ?)$ or $(?, r, t)$. [1].

Knowledge graph embedding (KGE) is a promising approach for predicting missing facts. It learns the embeddings of entities and relations in a low-dimensional vector space and defines score functions in this space to capture the connectivity patterns among the elements (entities and relations) of facts. For instance, TransE [4] represents relations as translations between two entities in Euclidean space, which can model the composition patterns. RotatE [32] represents entities as points in a complex space and relations as rotations, which is able to model symmetry/antisymmetry patterns. And HAKE [47] maps entities into a polar coordinate system, which can naturally reflect the hierarchy patterns. Although further works have successfully defined various score functions in specific vector space for effectively handling different subsets of patterns, due to the complexity of real-world knowledge graphs, the connectivity patterns between the entities and relations are also very complex. Therefore, it is difficult for traditional KGE methods to model all patterns solely through an explicit score function in a vector space.

To address these issues, some works attempt to design more complex score functions in specific vector spaces (e.g., non-Euclidean space, Complex space) to simultaneously capture multiple types of patterns. Other works try to combine different KGE models to capture more patterns. For example, [18] proposes to combine different knowledge graph embeddings through score concatenation to improve the performance in the KGC task. [39] proposes to combine the scores of different embedding models by using a weighted sum. A recent work SEA [13] introduces a query attention mechanism for the combination of the score functions of different KGE models. Although these methods can model more patterns compared to traditional KGE methods, the limited number of combination models still restricts their ability to capture more patterns and improve knowledge graph completion tasks. How to efficiently embed KGs and capture the variety of patterns in KGs remains a challenging problem.

Instead of designing complex score functions or combining different KGE models, this paper argues that plausible facts in a knowledge graph come from a distribution in the low-dimensional fact space. To this end, we propose a novel framework called Fact Embedding through Diffusion Model (FDM). FDM directly learns the distribution of plausible facts through Denoising Diffusion Probabilistic Models (DDPM) and casts the entity prediction task into the conditional fact generation task. Specifically, we concatenate the embeddings of elements in a fact as a whole and take it as input. Then, we introduce a Conditional Fact Denoiser to learn the reverse denoising diffusion process and generate the target fact embedding from noised data. A key challenge in the FDM is that diffusion processes typically operate in continuous space, while facts in KGs are inherently discrete. To address this gap, we propose a learnable Fact Embedding Module to map facts into vectors and perform diffusion directly in a continuous vector space. Additionally, to better guide the generation process in DDPM, we incorporate explicit conditional constraints into the reverse diffusion process and propose a Conditional Encoder to encode the known condition embeddings

---

[1] The problem $(?, r, t)$ is the same. So, this paper only discusses $(h, r, ?)$

and learn different connectivity patterns in them. Our contributions can be summarized as follows.

- We propose a novel framework called FDM, which directly learns the distribution of plausible facts through Denoising Diffusion Probabilistic Models (DDPM) and casts the entity prediction task into the conditional fact generation task. To the best of our knowledge, FDM is the first attempt to explore the potential of diffusion models for knowledge graph completion tasks.
- A learnable Fact Embedding Module is proposed to bridge the gap between continuous diffusion models and discrete facts in KGs. Furthermore, we propose a novel Conditional Fact Denoiser with constraints to learn the reverse diffusion process and generate the target fact and entity from noised data.
- Extensive experiments on four benchmark datasets demonstrate that FDM achieves superior performances in KGC tasks and significantly outperforms all types of state-of-the-art methods on three datasets. Especially on FB15k-237, FDM achieves a 16.8% relative improvement in MRR scores compared to the state-of-the-art methods.

## 2 RELATED WORK

**KG embedding methods:** Knowledge graph embedding aims to encode entities and relations into a continuous vector space, where the embeddings are required to preserve the connectivity patterns and semantic meaning of the original KG. The general intuition of these methods is to model and infer the connectivity patterns such as symmetry/antisymmetry, inversion, composition, and so on, between entities and relations based on observed facts. They [4, 6, 32, 43] focus on mapping entities and relations to a general or designed space (e.g., non-Euclidean space, Complex space) and defining a relation-dependent score function $f_r(h, t)$ in this space to model these patterns. For example, DistMult [43] represents each relation as a diagonal matrix. Its score function captures pairwise interaction between the same dimension of the head and tail embedding. Thus, DistMult treats symmetric relations well. RotatE [32], representing entities as points in a complex space and relations as rotations, can model relation patterns including symmetry/antisymmetry, inversion, and composition. GIE [6] learns spatial patterns interactively between the Euclidean, hyperbolic, and hyperspherical spaces. However, there is no single score function that can model all patterns. So, some works consider how to integrate different Knowledge Graph Embedding models to support various connectivity patterns. For instance, [42] has demonstrated that, under certain conditions, the ensemble model generated from the combination of multiple runs of low-dimensional embedding models of the same kind outperforms the corresponding individual high-dimensional embedding model. [13] utilizes an attention mechanism to combine the score function from several models in a unified one to incorporate patterns that are independently captured by each model. Although these methods can model more patterns compared to traditional KGE models, due to the complexity of the real KGs, there are numerous types of connectivity patterns, making it difficult to model them through limited combinations of models. In contrast, this paper argues that plausible facts (triplets), as the

fundamental units in Knowledge Graphs, should come from a distribution in the low-dimensional fact space. Although [37] tries to model triplet, it learns a triplet distributor for each triplet to transfer the information about entities and relations in different spaces, rather than modeling the distribution of the facts. In contrast, the proposed FDM learns the distribution of plausible facts through DDPM and casts the entity prediction task into the corresponding conditional fact generation task, without explicitly modeling the connectivity patterns between entities and relations.

**Diffusion Model:** The diffusion model utilizes diffusion processes to model the generation and defines data sampling as a gradual denoising process, recovering from a complete Gaussian noise. The forward process gradually adds Gaussian noise to the data according to a predefined noise schedule until the time step $T$. In recent years, the class of diffusion-based (or score-based) deep generative models has shown remarkable performance in modeling high-dimensional multi-modal distributions [15, 31], and demonstrated the ability to generate high-quality and diverse samples [11, 27] on several benchmark generation tasks in the field of computer vision [11]. To handle discrete data, previous works have explored text diffusion models in discrete state spaces, which define a corruption process for discrete data. [1] introduce the multinomial diffusion for character-level text generation, the forward categorical noise is applied through the Markov transition matrix. [1] generalize discrete text diffusion models by introducing the absorbing state ([MASK]). However, discrete diffusion models may have issues with scaling the one-hot row vectors, and they can only unconditionally generate text samples in discrete space. [21] and [12] propose a new language model diffusion on the continuous latent representations and connect the discrete space of texts with continuous space using different mapping functions. However, in this paper, we explore the potential of diffusion models to learn the distribution of plausible facts in knowledge graphs. Furthermore, the reverse process in traditional DDPM often utilizes neural networks, such as a UNet [11, 15] or transformer [12, 21] to parameterize the conditional distribution $p(x_{t-1}|x_t)$. However, knowledge graphs are often stored in the form of facts, which is different from images or text. Therefore, in this work, we propose a novel MLP-based Conditional Fact Denoiser (CFDenoiser) for facts in knowledge graphs to learn the reverse diffusion process.

## 3 METHODOLOGY

### 3.1 Problem Formulation

Let $\mathcal{G} = (\mathcal{V}, \mathcal{E})$ be an instance of a knowledge graph, where $\mathcal{V}$ is the set of nodes and $\mathcal{E}$ is the set of edges. Each edge $e$ has a relation type $r \in \mathcal{R}$. Our goal is to predict the missing entities in $\mathcal{G}$, i.e., given an incomplete fact $(h, r, ?)$, we aim to predict the missing tail entity $t$. Noticing that the problem $(?, r, t)$ is the same, this paper only discusses $(h, r, ?)$. The traditional KGE methods carry out entity prediction through a ranking procedure. Specifically, in order to predict the tail entity of an incomplete fact $(h, r, ?)$, it makes the prediction $\hat{t}$ by relation-dependent score function $f_r(h)$ in vector space. Then they rank the distance between $\hat{t}$ and each entity $t$ in KGs to get the answer. While the proposed FDM uses the Conditional Fact Denoiser with constraints to generate the target fact $X^\tau$ and get the corresponding tail entity $X^t$ conditioned on the

given head entity $X^h$ and relation $X^r$ in vector space. Finally, it does the same ranking process to get the final answer. The following sections will provide a detailed introduction to the architecture of FDM and its training objectives.

## 3.2 The FDM Architecture

Figure 1 illustrates the architecture of the FDM. From a high-level perspective, FDM can be divided into two stages: forward diffusion process and reverse diffusion process with conditional denoising. Specifically, the forward process gradually adds Gaussian noise to the known facts from a predefined noise schedule until time step $T$. Then, in the reverse diffusion process, FDM learns to model the Markov transition from Gaussian distribution to the distribution of plausible facts in the vector space through the Conditional Fact Denoiser. In the inference stage, we use the trained Conditional Fact Denoiser to perform the denoising diffusion process and generate the target fact embedding from noised data. Additionally, to better guide the reverse process, we incorporate explicit conditional constraints into the reverse diffusion process and propose a Conditional Encoder to encode the known condition embeddings and learn different connectivity patterns in them. Finally, we obtain the corresponding tail entity embedding. Next, we will provide a detailed introduction to these two processes.

**Forward Process.** To apply a continuous diffusion model to discrete facts $(h, r, t)$. We define a learnable Fact Embedding Module to map the fact into vector space. Fact Embedding Module includes two learnable embedding functions: $EMB^e$ and $EMB^r$, which both are linear layers. They map each entity and relation to vectors $X^e \in \mathbb{R}^e$, $X^r \in \mathbb{R}^r$. Next, for the fact $\tau = (h, r, t)$, Fact Embedding Module construct the corresponding fact embedding $X^\tau$ as follows: $X^\tau = [X^h; X^r; X^t] \in \mathbb{R}^{2 \times r + e}$, where $[\cdot; \cdot]$ indicates concatenation on the sequence length dimension and $X^h, X^r, X^t$ are the output of two embedding functions applied to $h, r, t$. Our goal is to predict the $X^\tau$ condition on head entity embedding $X^h$ and relation embedding $X^r$. Finally, we get the tail entity embedding $X^t$ from the $X^\tau$. So, during the training stage, $q(X^\tau | X^h, X^r)$ is the unknown target fact distribution. To train the FDM, we define the forward diffusion process which maps the fact embedding $X^\tau$ to pure noise by gradually adding Gaussian noise at each time step $T_t = i$ until at diffusion step $T_t = T$. Each transition $X^\tau_{T_t-1} \rightarrow X^\tau_{T_t}$ is parametrized by:

$$q(X^\tau_{T_t} | X^\tau_{T_t-1}) = \mathcal{N}(X^\tau_{T_t}; \sqrt{1 - \beta_{T_t}} X^\tau_{T_t-1}, \beta_{T_t} I) \quad (1)$$

where $\{\beta_{T_t}\}_{T_t=1}^T$ are forward process variances. This parametrization of the forward process $q$ contains no trainable parameters and allows us to define a training objective.

**Reverse Process.** In the second stage, FDM defines the conditional reverse diffusion process $p(X^\tau_{T_t-1} | X^\tau_{T_t}, X^h, X^r)$ which performs iterative denoising from pure Gaussian noise to generate target fact $X^\tau$ conditioned on the embedding of known entity $X^h$ and relation $X^r$. Then, the transition between two nearby latent variables is denoted by:

$$p_\theta(X^\tau_{T_t-1} | X^\tau_{T_t}, X^h, X^r) =$$
$$\mathcal{N}(X^\tau_{T_t-1}; \mu_\theta(X^\tau_{T_t}, T_t, X^h, X^r), \sigma^2_{T_t} I) \quad (2)$$

where $\sigma_{T_t}$ is the constant variance following [15], $\mu_\theta$ is the mean of the Gaussian distribution computed by a denoiser, and $\theta$ is the parameters of the neural network. As shown in [15], we can reparameterize the mean to make the neural network learn the added noise at time step $T_t$ instead. In this way, $\mu_\theta$ can be reparameterized as follows:

$$\mu_\theta(X^\tau_{T_t}, T_t, X^h, X^r) = \frac{1}{\sqrt{\alpha_{T_t}}} \left( X^\tau_{T_t} - \frac{\beta_{T_t}}{\sqrt{1 - \bar{\alpha}_{T_t}}} \epsilon_\theta(X^\tau_{T_t}, T_t, X^h, X^r) \right) \quad (3)$$

Where $T_t$ is the time step, $\{\beta_{T_t}\}_{T_t=1}^T$ are forward process variances, $\alpha_{T_t} = 1 - \beta_{T_t}$, and $\bar{\alpha}_{T_t} = \prod_{s-1}^{T_t} \alpha_s$. $\epsilon_\theta(X^\tau_{T_t}, T_t, X^h, X^r)$ is the neural network to predict the added noise conditioned on known condition embeddings at time step $T_t$. The $\epsilon_\theta(X^\tau_{T_t}, T_t, X^h, X^r)$ is called Conditional Fact Denoiser (CFDenoiser) and we will introduce it in the next section.

Additionally, unlike the traditional conditional reverse diffusion process, FDM incorporates explicit conditional constraints (the known entities and relations) into the reverse diffusion process, aiming to ensure that the generated target fact embeddings satisfy the existing constraints as closely as possible. Specifically, for the tail entity prediction, i.e., $(h, r, ?)$, the embeddings of the conditions $h$ and $r$ not only serve as the classifier-free diffusion guidance in equation 3 but also ensure that at each step $p_\theta(X^\tau_{T_t-1} | X^\tau_{T_t}, X^h, X^r)$ of generating the target fact embedding $X^\tau_{T_t}$, the corresponding embeddings of the head entity and relation $X^{hr}_{T_t} = [X^h_{T_t}; X^r_{T_t}]$ in fact embedding are as consistent as possible with the known embeddings of the head entity and relation $X^{hr} = [X^h; X^r]$. Finally, we take the $F$ as the objective function and take gradient descent steps $s = 1, 2, \cdots, Gradstep$ at each denoising step:

$$F(X^{hr}_{T_t}, X^{hr}) = ||X^{hr}_{T_t} - X^{hr}||^2_2 \quad (4)$$

$$X^\tau_{T_t} = X^\tau_{T_t} - \eta \nabla_{X^{hr}_{T_t}} F(X^{hr}_{T_t}, X^{hr}) \quad (5)$$

Where $F$ is the differentiable objective function (squared $L2$ norm), $\eta$ is the fixed learning rate. Since $X^{hr}_{T_t}$ is part of $X^\tau_{T_t}$, the gradient of $X^t_{T_t}$ in fact embedding is approximated by averaging the gradient of $X^{hr}_{T_t}$. The number of gradient descent steps $Gradstep$ is a hyperparameter that will be discussed in the next section.

## 3.3 Conditional Fact Denoiser

In the reverse process of DDPM, the most important thing is how to design a suitable denoising model for the data. Currently, most existing denoising models in DDPM are mainly designed for image or text data. However, knowledge graphs are often stored in the form of facts $(h, r, t)$, which have shorter lengths and less obvious long-range dependencies. Therefore, we propose a simple and efficient MLP-based Conditional Fact Denoiser (CFDenoiser) for handling knowledge graphs instead of using transformers as

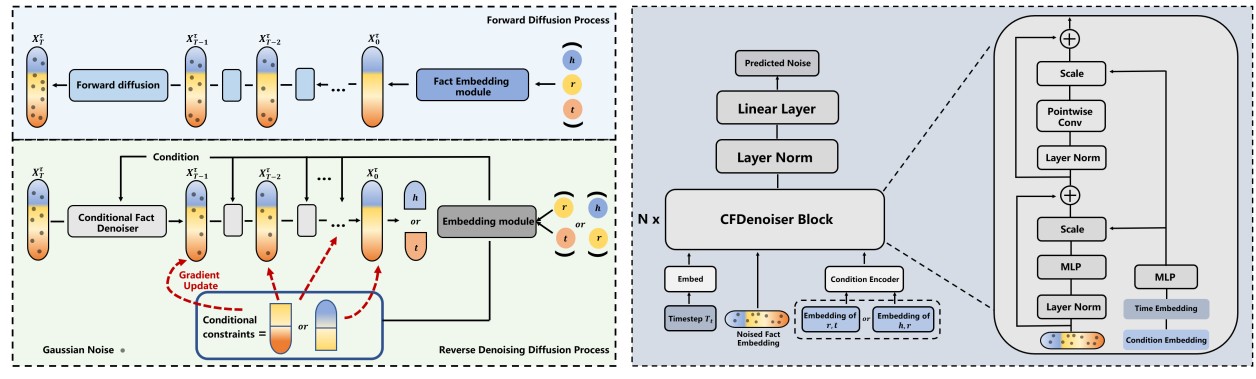

**(a) FDM Architecther**  **(b) Conditional Fact Denoiser**

Figure 1: (a) Architecture of FDM. It consists of a forward diffusion process and a reverse diffusion process modeled by a Conditional Fact Denoiser. (b) Architecture of Conditional Fact Denoiser (CFDenoiser).

the backbone. We conduct ablation experiments in the following sections to demonstrate that transformers are not suitable for the KG data. The architecture of CFDenoiser is illustrated in Figure 1(b). Formally, the architecture of CFDenoiser can be described as follows:

$$X^c = ConditionEncoder(X^h, X^r) \tag{6}$$

$$E = CFDenoiserBlock(X^\tau_{T_t}, X^T_{T_t}, X^c) \tag{7}$$

$$\epsilon = LinearLayer(LN(E)) \tag{8}$$

where $X^h$ and $X^r$ represent the embeddings of entity $h$ and relation $r$, respectively. $X^c$ is the final condition embedding calculated by Condition Encoder. $X^\tau_{T_t}$ is the noised fact embedding at step $T_t$ and $X^T_{T_t}$ denotes the timestep embedding at step $T_t$. $E$ is intermediate feature calculated by the CFDenoiser block, and $\epsilon$ is the noise predicted by CFDenoiser. Next, we will introduce the Condition Encoder and the CFDenoiser block.

**Condition Encoder.** As mentioned above, when generating the target facts from noise in fact space, CFDenoiser uses the known embeddings $X^h, X^r$ to guide the generation process in DDPM. After the vectorization of conditions, most previous works [16, 21] concatenate the different conditional embeddings simply and use them as the final control condition. However, for facts in KGs, the entities and relations usually have rich patterns and are not independent of each other. Inspired by the existing neural network-based model [10] in modeling the patterns among entities and relations in KGs, the Condition Encoder utilizes the linear layer in vector space to learn different patterns, which can better guide the generation process. The Condition Encoder can be represented as follows:

$$ConditionEncoder(X^h, X^r) = LinearLayer(X^h \oplus X^r) \tag{9}$$

where $\oplus$ denotes the Hadamard product in complex vector space and the addition in real vector space. In FB15k-237, we use the addition. In WN18RR, Kinship, and UMLS, we use the Hadamard product. In the subsequent experiments, we conduct ablation studies to demonstrate that the performance of simply concatenate conditional embeddings is far inferior compared to using the Condition Encoder.

**CFDenoiser Block.** Inspired by the success of the transformer encoder [36] in the graph data domain [17], the CFDenoiser Block adopts a similar architecture. It consists of alternating layers of MLP. Layernorm (LN) is applied before every layer and residual connections are employed around each of the sub-layers [38]. As mentioned above, the form of facts $(h, r, t)$ is simple with a short length, and their long-range dependencies are not evident. CFDenoiser utilizes simple MLP layers instead of multiheaded self-attention layers. Furthermore, in order to make full use of the conditional embeddings for guiding the generation process, we regress dimension-wise scaling parameters $\alpha$ which are applied immediately prior to every residual connections [25] within the sub-layers as shown in Figure 1(b).

### 3.4 Training and Inference

Since negative sampling has been proven quite effective for both learning knowledge graph embeddings [34] and word embeddings [24]. we use a loss function similar to the negative sampling loss [24]:

$$L = -log\, \sigma(\gamma - d_1(X^\tau, Denoise(X^\tau))) -$$
$$\sum_{i=1}^n \frac{1}{k} log\, \sigma(d_1(X^\tau, Denoise(X^{\bar{\tau}_i})) - \gamma) \tag{10}$$

where $\gamma$ is a fixed margin, $\sigma$ is the sigmoid function, and the $d_1$ is the $L1$ distance. The predicted noise and the final denoised results can convert to each other [15] by $Denoise(X^\tau) = \frac{1}{\sqrt{\bar{\alpha}_{T_t}}}X^\tau_{T_t} - \sqrt{\frac{1}{\bar{\alpha}_{T_t}} - 1}\, \epsilon_\theta(X^\tau_{T_t}, T_t, X^h, X^r)$. And $\bar{\tau}_i = (h, r, \bar{t}_i)$ is the $i$-th negative fact for the tail entity prediction on the positive sample $\tau = (h, r, t)$. For the head prediction, we replace the corresponding $\bar{h}_i$.

In the inference stage, when predicting $(h, r, ?)$, FDM uses the trained CFDenoiser and corresponding conditions (known embedding of the entity $X^h$ and relation $X^r$) to iteratively denoise from pure Gaussian noise. Subsequently, it predicts target fact embedding $X^\tau$ and obtains the prediction of tail entity $X^t_{pre}$. Finally, we rank the distance between $X^t_{pre}$ and each entity $X^t$ in KGs to get the final prediction $t$. The training and inference algorithms of FDM are presented in Algorithm 1 and Algorithm 2 respectively.

---

**Algorithm 1:** Training Stage

**Input**: $\tau = (h, r, t)$, $\bar{\tau} = (h, r, \bar{t})$;
**Parameters**: $EMB^e$, $EMB^r$, $CFDenoiser$: $\epsilon_\theta$;
**repeat**
    Calculate $X^\tau$ and $X^{\bar{\tau}}$ by Fact Embedding Module.
    $T_t \sim \text{Uniform}\ (\{1, \cdots, T\})$;
    $\epsilon \sim \mathcal{N}(\mathbf{0}, \mathbf{I})$;
    $X^\tau_{T_t} = \sqrt{\bar{\alpha}_{T_t}} X^\tau + \sqrt{1 - \bar{\alpha}_{T_t}} \epsilon$;
    $Denoise(X^\tau) = \frac{1}{\sqrt{\bar{\alpha}_{T_t}}} X^\tau_{T_t} - \sqrt{\frac{1}{\bar{\alpha}_{T_t}} - 1} \epsilon_\theta(X^\tau_{T_t}, T_t, X^h, X^r)$;
    Take the gradient descent step on:
    $L = -\log \sigma(\gamma - d_1(X^\tau, Denoise(X^\tau))) -$
    $\sum_{i=1}^n \frac{1}{k} \log \sigma(d_1(X^\tau, Denoise(X^{\bar{\tau}_i})) - \gamma)$;
**until** *converged*;

---

**Algorithm 2:** Inference Stage

**Input**: Incomplete fact $(h, r, ?)$;
**Output**: Predicted target fact embedding $X^\tau$;
$X^\tau_{T_t} \sim \mathcal{N}(\mathbf{0}, \mathbf{I})$;
$X^h \leftarrow EMB^e(h)$, $X^r \leftarrow EMB^r(r)$;
**for** $T_t = T, \cdots, 1$ **do**
    $\mathbf{z} \sim \mathcal{N}(\mathbf{0}, \mathbf{I})$ if $T_t > 1$, else $\mathbf{z} = \mathbf{0}$;
    $X^\tau_{T_t-1} = \frac{1}{\sqrt{\alpha_{T_t}}}(X^\tau_{T_t} - \frac{1-\alpha_{T_t}}{\sqrt{1-\bar{\alpha}_{T_t}}} \epsilon_\theta(X^\tau_{T_t}, T_t, X^h, X^r)) + \sigma_{T_t} \mathbf{z}$;
    **for** $s = 1, 2, \cdots, Gradstep$ **do**
        $X^\tau_{T_t-1} = X^\tau_{T_t-1} - \eta \nabla_{X^{hr}_{T_t}} F(X^{hr}_{T_t}, X^{hr})$
    **end**
**end**
**return** $X^\tau$

---

## 4 EXPERIMENT

### 4.1 Experiment Setup

**Datasets**: We select four typical KGC datasets for evaluation, including FB15k-237 [33], WN18RR [10], Kinship and UMLS. For Kinship and UMLS, we use the datasets division in [26]. Statistics of datasets can be found in the table 1.

**Baselines**: We compared with the four types of KGC methods following [8]. **Knowledge graph embedding methods**: TransE [4], DualE [5], DistMult [43], ComplEx [34], ComplEx-N3 [19], TuckER [2], ConvE [10], RotatE [32], HAKE [47], GIE [6], ATTH [7], SEA [13], AnKGE-HAKE [45] and TDN [37]. **Path-based methods**: RNN-Logic [26], NeuralLP[44], DRUM [28], PathRank [20], MINERVA [9], and M-Walk[30]. **Graph neural networks methods**: NBFNet [48], COMPGCN [35], HKGN [23], DRGI [22], RGCN [29] **Instance-based learning methods**: IBLE and CIBLE [8]. For the FB15K-237, WN18RR, Kinship, and UMLS, we cite results in [8] and [13] for comparison.

**Evaluation Protocols**: For each test fact $(h, r, t)$, we construct two queries: $(h, r, ?)$ and $(?, r, t)$, with the answers $t$ and $h$. The Mean Rank (MR), Mean Reciprocal Rank (MRR), and H@N are reported under the filtered setting [32], in line with previous research. Higher MRR and Hit@N indicate a better performance

**Implementation details**: Our model is trained on 2 Nvidia A100

GPU. We describe the hyper-parameter, architectures, and more experimental details in appendix A and appendix B.

**Table 1: Statistics of datasets. The symbols Ent and Rel denote the number of entities and relations respectively.**

| Dataset | Ent | Rel | Train | Validation | Test |
|---|---|---|---|---|---|
| **FB15k-237** | 14,541 | 237 | 272,115 | 17,535 | 20,466 |
| **WN18RR** | 40,943 | 11 | 86,853 | 3,034 | 3,134 |
| **Kinship** | 104 | 25 | 3,206 | 2,137 | 5,343 |
| **UMLS** | 135 | 46 | 1,959 | 1,306 | 3,246 |

### 4.2 Main Results

The main results are presented in Tables 2, and Table 3. We categorize the existing KGC methods into two main groups, non-embedding methods (including Path-based methods, graph neural networks methods, and Instance-based learning methods) and embedding methods. The non-embedding methods are listed in the upper section of the table, while the embedding methods are listed in the lower section of the table. Our observations based on the results are as follows. First, compared to embedding methods, FDM shows remarkable improvement across all metrics on four datasets. Specifically, it achieves a 10% (25.9% relative), 0.6% (1.2% relative), 5.7% (7.3% relative), and 6.2% (7.2% relative) increase in MRR scores over the best embedding models on the FB15k-237, WN18RR, Kinship, and UMLS respectively. Second, compared to non-embedding methods, FDM achieves better results for all metrics on the FB15k-237, Kinship, and UMLS datasets. On the WN18RR, FDM retains its superiority over other non-embedding methods except for the NBFNet. Specifically, FDM achieves a 7% (16.8% relative), 7.7% (10.1% relative), and 8% (9.5% relative) increase in MRR scores over the best non-embedding models on the FB15k-237, Kinship, and UMLS datasets, respectively. In conclusion, these results illustrate that by modeling the distribution of plausible facts in a low-dimensional fact space and converting the entity prediction into conditional fact generation, FDM can improve the performance of embedding methods greatly. Furthermore, we notice that the performance improvement of FDM on the WN18RR is not significant. We analyze that it is caused by the higher entity-to-relation ratio (the number of entities/ the number of relations) on WN18RR(40943/11) than the other three datasets: FB15k-237 (14541/237), UMLS (135/46), and Kinship (104/25). The larger the entity-to-relation ratio implies a more complex target fact distribution and connectivity patterns, which are more difficult to model by FDM.

Next, we focus on complex multi-relation scenarios, especially for the issue of 1-N, N-1, and N-N relations (as shown in Table 4). This is because, in these complex relation scenarios, the patterns between entities and relations become more complicated and increasingly challenging to model. We present the experimental results on different relation types following the [40]. We choose the FB15k-237 owing to its abundant multi-relations and denser graph structure. Then, we compare FDM with TransE [4], RotatE [32], COMPGCN [35] and NBFNet [48]. It is observed that FDM shows a greater relative improvement in the 1-N, N-1, and N-N types, which illustrates that compared to other KGE models and GNN-based

**Table 2: Entity prediction results on FB15k-237 and WN18RR. The best results are in bold and the second best results are underlined.**

| Model | FB15k-237 | | | | WN18RR | | | |
|---|---|---|---|---|---|---|---|---|
| | MRR ↑ | H@1 ↑ | H@3 ↑ | H@10 ↑ | MRR ↑ | H@1 ↑ | H@3 ↑ | H@10 ↑ |
| **Non-embedding methods** | | | | | | | | |
| PathRank [20] | 0.087 | 7.4 | 9.2 | 11.2 | 0.189 | 17.1 | 20.0 | 22.5 |
| NeuralLP [44] | 0.237 | 17.3 | 25.9 | 36.1 | 0.381 | 36.8 | 38.6 | 40.8 |
| DRUM [28] | 0.238 | 17.4 | 26.1 | 36.4 | 0.382 | 36.9 | 38.8 | 41.0 |
| RNNLogic [26] | 0.344 | 25.2 | 38.0 | 53.0 | 0.483 | 44.6 | 49.7 | 55.8 |
| RGCN [29] | 0.273 | 18.2 | 30.3 | 45.6 | 0.402 | 34.5 | 43.7 | 49.4 |
| COMPGCN [35] | 0.355 | 26.4 | 39.0 | 53.5 | 0.479 | 44.3 | 49.4 | 54.6 |
| HKGN [23] | 0.365 | 27.2 | 40.2 | 55.2 | 0.487 | 44.8 | 50.5 | 56.1 |
| NBFNet [48] | 0.415 | 32.1 | 45.4 | 59.9 | **0.551** | **49.7** | **57.3** | **66.6** |
| CIBLE [8] | 0.341 | 24.5 | 37.7 | 53.7 | 0.490 | 44.6 | 50.7 | 57.5 |
| **Embedding methods** | | | | | | | | |
| TransE [4] | 0.294 | - | - | 46.5 | 0.226 | - | - | 50.1 |
| ConvE [10] | 0.325 | 23.7 | 35.6 | 50.1 | 0.430 | 40.0 | 44.0 | 52.0 |
| RotatE [32] | 0.338 | 24.1 | 37.5 | 53.3 | 0.476 | 42.8 | 49.2 | 57.1 |
| ATTH [7] | 0.348 | 25.2 | 38.4 | 54.0 | 0.486 | 44.3 | 49.9 | 57.3 |
| DualE [5] | 0.365 | 26.8 | 40.0 | 55.9 | 0.492 | 44.4 | 51.3 | 58.4 |
| GIE [6] | 0.362 | 27.1 | 40.1 | 55.2 | 0.491 | 45.2 | 50.5 | 57.5 |
| SEA [13] | 0.360 | 26.4 | 39.8 | 54.9 | 0.500 | 45.4 | 51.8 | 59.1 |
| AnKGE-HAKE [45] | 0.385 | 28.8 | 42.8 | 57.2 | 0.500 | 45.4 | 51.5 | 58.7 |
| TDN [37] | 0.350 | 26.3 | 39.5 | 54.6 | 0.481 | 43.9 | 50.2 | 48.1 |
| **FDM(Ours)** | **0.485** | **38.6** | **52.9** | **68.1** | 0.506 | 45.6 | 51.8 | 59.2 |

**Table 3: Entity prediction results on Kinship and UMLS. The best results are in bold and the second best results are underlined.**

| Model | Kinship | | | | UMLS | | | |
|---|---|---|---|---|---|---|---|---|
| | MRR ↑ | H@1 ↑ | H@3 ↑ | H@10 ↑ | MRR ↑ | H@1 ↑ | H@3 ↑ | H@10 ↑ |
| **Non-embedding methods** | | | | | | | | |
| PathRank [20] | 0.369 | 27.2 | 41.6 | 67.3 | 0.197 | 14.8 | 21.4 | 25.2 |
| NeuralLP [44] | 0.302 | 16.7 | 33.9 | 59.6 | 0.483 | 33.2 | 56.3 | 77.5 |
| MINERVA [9] | 0.401 | 23.5 | 46.7 | 76.6 | 0.564 | 42.6 | 65.8 | 81.4 |
| DRUM [28] | 0.334 | 18.3 | 37.8 | 67.5 | 0.548 | 35.8 | 69.9 | 85.4 |
| RNNLogic [26] | 0.722 | 59.8 | 81.4 | 94.9 | 0.842 | 77.2 | 89.1 | 96.5 |
| DRGI [22] | 0.760 | 58.6 | 84.3 | 95.9 | 0.820 | 77.1 | 83.8 | 96.1 |
| NBFNet [48] | 0.606 | 43.5 | 72.5 | 93.7 | 0.778 | 68.8 | 84.0 | 93.8 |
| IBLE [8] | 0.615 | 45.9 | 71.7 | 92.8 | 0.816 | 71.7 | 90.0 | 96.1 |
| CIBLE [8] | 0.728 | 60.3 | 82.0 | 95.6 | 0.831 | 74.9 | 89.7 | 97.0 |
| **Embedding methods** | | | | | | | | |
| DistMult [43] | 0.241 | 15.5 | 26.3 | 41.9 | 0.430 | 39.0 | 44.0 | 49.0 |
| ComplEx [34] | 0.247 | 15.8 | 27.5 | 42.8 | 0.440 | 41.0 | 46.0 | 51.0 |
| ComplEx-N3 [19] | 0.605 | 43.7 | 71.0 | 92.1 | 0.791 | 68.9 | 87.3 | 95.7 |
| TuckER [2] | 0.603 | 46.2 | 69.8 | 86.3 | 0.732 | 62.5 | 81.2 | 90.9 |
| RotatE [32] | 0.651 | 50.4 | 75.5 | 93.2 | 0.744 | 63.6 | 82.2 | 93.9 |
| ConvE [10] | 0.685 | 55.2 | 78.6 | 92.8 | 0.756 | 69.7 | 80.7 | 91.9 |
| TDN [37] | 0.780 | 67.7 | 86.7 | 96.5 | 0.860 | 82.4 | 87.3 | 96.8 |
| **FDM(Ours)** | **0.837** | **76.1** | **89.7** | **96.8** | **0.922** | **89.3** | **94.4** | **97.0** |

**Table 4: Entity prediction results by relation category [40] on FB15k-237 dataset. The best results are in bold and the second best results are underlined.**

| | | TransE | | RotatE | | COMPGCN | | NBFNet | | FDM | |
|---|---|---|---|---|---|---|---|---|---|---|---|
| | | MRR↑ | H@10↑ | MRR↑ | H@10↑ | MRR↑ | H@10↑ | MRR↑ | H@10↑ | MRR↑ | H@10↑ |
| **Head Pred** | 1-1 | 0.498 | - | 0.487 | 0.593 | 0.457 | 0.604 | **0.578** | - | 0.569 | **0.708** |
| | 1-N | 0.079 | - | 0.081 | 0.174 | 0.112 | 0.190 | 0.165 | - | **0.203** | **0.370** |
| | N-1 | 0.455 | - | 0.467 | 0.674 | 0.471 | 0.656 | 0.499 | - | **0.559** | **0.744** |
| | N-N | 0.224 | - | 0.234 | 0.476 | 0.275 | 0.474 | 0.348 | - | **0.423** | **0.657** |
| **Tail Pred** | 1-1 | 0.488 | - | 0.484 | 0.578 | 0.453 | 0.589 | **0.600** | - | 0.519 | **0.609** |
| | 1-N | 0.744 | - | 0.747 | 0.674 | 0.779 | 0.885 | 0.790 | - | **0.826** | **0.904** |
| | N-1 | 0.071 | - | 0.070 | 0.138 | 0.076 | 0.151 | 0.122 | - | **0.167** | **0.301** |
| | N-N | 0.330 | - | 0.338 | 0.608 | 0.395 | 0.616 | 0.456 | - | **0.543** | **0.757** |

**Table 5: Ablation on Condition Encoder in the CFDenoiser in FB15k-237 and Kinship.**

| Model | FB15k-237 | | Kinship | |
|---|---|---|---|---|
| | MRR↑ | H@1↑ | MRR↑ | H@1↑ |
| **FDM w/o Condition Encoder** | 0.457 | 35.3 | 0.781 | 68.3 |
| **FDM w/ Condition Encoder** | **0.485** | **38.6** | **0.837** | **76.1** |

**Table 6: Ablation on MLP-based architecture of the CFDenoiser in FB15k-237 and Kinship.**

| Model | FB15k-237 | | Kinship | |
|---|---|---|---|---|
| | MRR↑ | H@1↑ | MRR↑ | H@1↑ |
| **FDM w/ transformer-based** | 0.318 | 22.3 | 0.601 | 44.2 |
| **FDM w/ MLP-based** | **0.485** | **38.6** | **0.837** | **76.1** |

models, FDM performs better in handling complex relationships and excels at modeling patterns in KGs under complex relation scenarios.

## 4.3 Ablation Study

**MLP-based architecture of the CFDenoiser.** First, we conduct an ablation study to prove that an MLP-based architecture is more suitable for KGs in the conditional reverse process. We replace the MLP layers in the CFDenoiser block with a two-layer transformer and then observe the performance of FDM on the FB15k-237 and Kinship datasets. The results in Table 6 show that the performance of FDM decreases by nearly 35% when using transformer layers in the CFDenoiser block, indicating that the MLP layers are more suitable for modeling simple-form facts in knowledge graphs. On the contrary, a more complex transformer is not suitable for modeling this relatively simple form of KG data.

**Condition Encoder in the CFDenoiser.** Next, In this subsection, we analyze the necessity of the Condition Encoder in the CFDenoiser on FB15k-237 and Kinship. We compare the results between using the Condition Encoder and directly concatenating the conditions embedding. The contributions of the Condition Encoder are summarized in Table 5. It can be observed that without the Condition Encoder, the performance of FDM drops by about 6% on average, illustrating that the entities and relations in KGs are not independent of each other. Simply concatenating their embeddings cannot capture the patterns in the facts. By using the Condition Encoder to learn different patterns, which can better guide the generation in the reverse diffusion process.

**hyper-parameters.** Finally, we conduct ablation experiments on hyper-parameters, including the number of the CFDenoiser block, the hidden size of the MLP layer in the CFDenoiser block, and the number of gradient descent steps (*Gradstep*) when applying conditional constraints. Figure 2 reports the results on UMLS and Kinship in terms of MRR scores. Figure 2(a) indicates the importance of shallow and proper numbers of CFDenoiser blocks (in this case, 2). Since the inputs of CFDenoiser are relatively simple, using a very deep network may result in overfitting and decrease the performance. Figure 2(b) explores the impact of the hidden size of the MLP layer in the CFDenoiser block on the model. The results demonstrate that increasing the hidden size helps improve the performance of the model, but too large sizes (e.g., 2200) will lead to a decrease in performance. Lastly, as mentioned earlier, we hope that the generated fact embeddings can satisfy the conditional constraints and we achieve this goal through gradient descent at each denoising step. Figure 2(c) studies the influence of the number of *Gradstep* in the reverse diffusion process. It indicates that a large number of *Gradstep* does not lead to a significant improvement in the final result but is more time-consuming. On the other hand, an appropriate number of *Gradstep* (e.g., 10) can help the generated results meet the constraints and exhibit a noticeable improvement efficiently compared to unconstrained results.

## 4.4 Case Study

Finally, we explore the better performance of FDM on real KGs. We provide an example from FB15k-237. The fact to be predicted

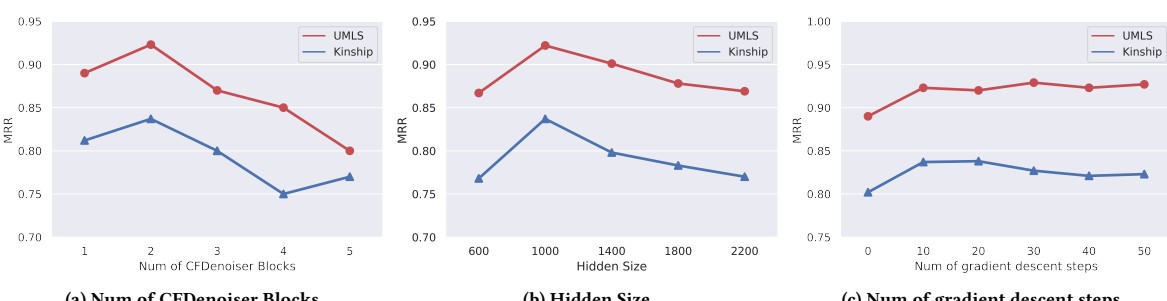

**Figure 2: hyper-parameters analysis on UMLS and Kinship (MRR).**

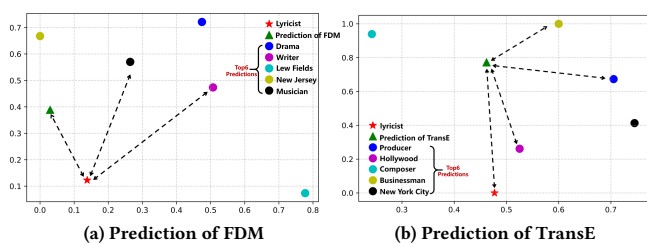

**Figure 3: T-SNE visualization to the predictions of ( Dorothy Fields, /people/person/profession, ? ) calculated by FDM and TransE.**

is ( Dorothy Fields, /people/person/profession, ? ). For this question, the answer is the lyricist. We visualize the prediction embeddings calculated by the FDM and the corresponding six entities closest to the prediction in Figure 3(a). The prediction results calculated by trained TransE are shown in Figure 3(b). From this visualization, we can observe that the embedding predicted by TransE: $Pred_{TransE}(t) = h + r$ is closer to other entities such as producer and composer rather than lyricist. While the prediction of the FDM is closest to the lyricist. This result illustrates that FDM can effectively model the distribution of target fact and make a more precise prediction compared to traditional KGE methods like TransE.

## 5 CONCLUSION

In this paper, we propose a novel framework called Fact Embedding through Diffusion Model (FDM) to address knowledge graph completion tasks. Different from most existing KGE models typically mapping entities and relations to a designed space and defining a relation-dependent score function to model connectivity patterns, we argue that plausible facts in a knowledge graph come from a distribution in the low-dimensional fact space. The FDM is proposed to directly learn the distribution of plausible facts from a known knowledge graph and cast the entity prediction task into the conditional fact generation task. Extensive experiments demonstrate that FDM significantly outperforms existing state-of-the-art methods on benchmark datasets for KGC tasks.

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

# APPENDIX

# A IMPLEMENTATION DETAILS

We implement our code with Pytorch. For our method, we finetune the hyper-parameters including batch size ranging from $\{128, 256, 512, 1024\}$, negative sampling size ranging from $\{128, 256, 512\}$, learning rate ranging from $\{1e^{-4}, 1e^{-5}, 5e^{-5}, 8e^{-5}\}$, the margin $\gamma$ ranging from $\{10, 14, 18, 22, 26, 30\}$, the iteration of gradient descent steps $Gradstep$ ranging from $\{10, 20, 30, 40, 50\}$, the number of the CFDenoiser block ranging from $\{1, 2, 3, 4, 5, 6\}$ and the hidden size of MLP layer in the CFDenoiser block ranging from $\{200, 400, 800, 1200, 1600, 2000, 2400\}$. We list the best hyper-parameter setting for each dataset in Table 7. We train the FDM on 2 Nvidia Quadro RTX 8000 GPUs with standard data parallelism. We select the best checkpoint on its performance on the validation set and the selection criteria is MRR. During inference, we generate the output 20 times and select the best result.

# B MORE EXPERIMENTAL RESULTS

In this section, we provide additional experimental results that are not included in the main text due to space limitations. First, table 8 shows the results of FDM under different diffusion time steps. The results indicate that the optimal diffusion time step is 1000. At this step, the proposed FDM demonstrates high prediction performance and time efficiency. Increasing the number of diffusion steps is highly time-consuming and does not significantly improve the performance of FDM. Decreasing the number of steps to 500 results in a decrease in the performance. Therefore, selecting an efficient diffusion time step is crucial for FDM.

Second, table 9 and table 10 show the performance of all metrics during ablation experiments on the Condition Encoder of the CFDenoiser and the MLP-based architecture of the CFDenoiser respectively. From the table 9, it can be observed that without the Condition Encoder, the performance of FDM drops by about 6% on average, illustrating that the entities and relations in KGs are not independent of each other. Simply concatenating their embeddings cannot capture the patterns in the facts. The Condition Encoder can better guide the generation process. Table 10 shows that the performance of FDM decreases by nearly 35% when using transformer layers in the CFDenoiser block, indicating that the MLP layers are more suitable for modeling simple-form facts in knowledge graphs.

**Table 7: The best hyper-parameter configurations of FDM on different datasets.**

| Hyper-parameter | | FB15k-237 | WN18RR | Kinship | UMLS |
|---|---|---|---|---|---|
| **CFDenoiser** | number of blocks | 3 | 2 | 2 | 2 |
| | hidden size | 1600 | 500 | 2000 | 2000 |
| **Batch** | positive | 512 | 256 | 512 | 256 |
| | negative | 256 | 128 | 256 | 128 |
| **Learning** | optimizer | Adam | Adam | Adam | Adam |
| | learning rate | $8e^{-5}$ | $8e^{-5}$ | $5e^{-5}$ | $8e^{-5}$ |
| | gamma | 27 | 46 | 14 | 10 |
| | *Gradstep* | 10 | 10 | 10 | 10 |

**Table 8: Results of different number of diffusion timesteps on FB15k-237.**

| Diffusion timesteps | MRR | H@1 | H@3 | H@10 |
|---|---|---|---|---|
| 500 | 0.462 | 37.1 | 50.9 | 67.2 |
| **1000** | 0.485 | **38.6** | **52.9** | 68.1 |
| 2000 | 0.481 | 38.4 | 52.1 | 67.7 |
| 3000 | 0.486 | 38.5 | 52.5 | 68.9 |
| 4000 | **0.488** | 38.6 | 52.7 | **69.0** |

**Table 9: Ablation on Condition Encoder of the CFDenoiser in FB15k-237 and Kinship.**

| Model | FB15k-237 | | | | Kinship | | | |
|---|---|---|---|---|---|---|---|---|
| | MRR | H@1 | H@3 | H@10 | MRR | H@1 | H@3 | H@10 |
| **FDM w/o Condition Encoder** | 0.457 | 35.3 | 49.4 | 64.8 | 0.781 | 68.3 | 84.3 | 95.9 |
| **FDM w/ Condition Encoder** | **0.485** | **38.6** | **52.9** | **68.1** | **0.837** | **76.1** | **89.7** | **96.8** |

**Table 10: Ablation on MLP-based architecture of the CFDenoiser in FB15k-237 and Kinship.**

| Model | FB15k-237 | | | | Kinship | | | |
|---|---|---|---|---|---|---|---|---|
| | MRR | H@1 | H@3 | H@10 | MRR | H@1 | H@3 | H@10 |
| **FDM w/ transformer-based** | 0.318 | 22.3 | 32.7 | 46.3 | 0.601 | 44.2 | 69.5 | 92.3 |
| **FDM w/ MLP-based** | **0.485** | **38.6** | **52.9** | **68.1** | **0.837** | **76.1** | **89.7** | **96.8** |

