# OpenReview forum: "Fact Embedding through Diffusion Model for Knowledge Graph Completion"
_ACM.org/TheWebConf/2024/Conference — TheWebConf24_

### Official Review · Reviewer_qnKh · 2023-11-21

**Novelty:** 5
**Technical Quality:** 4

**Review:**

The paper introduces the recent popular Denoising Diffusion Probabilistic Models (DDPM) into the KGE field, transforming the entity prediction task into a conditional fact generation task. Departing from traditional methods of defining a score function to capture the intricate connectivity patterns, the proposed FDM method learns the distribution of plausible facts directly from the knowledge graphs. This approach employs a Conditional Fact Denoiser to learn the reverse denoising diffusion process, thereby generating the target fact embedding from noised data. Experimental results on four datasets demonstrate the effectiveness of FDM.

Strengths:
1. This paper offers a fresh perspective by learning the distribution of plausible facts rather than relying on a score function. It is a creative use of diffusion models in the context of KGE, suggesting a novel direction for future research in this field.

2. A broad range of baselines are compared in experiments including both embedding-based and non-embedding-based recent models. The notable improvement on the FB15k-237 dataset, strongly supports the proposed framework's superiority over existing methods.

3. The manuscript is well-organized and easy to follow.

Weaknesses:
1. The paper does not fully discuss the computational complexity and scalability of FDM, particularly when applied to large-scale knowledge graphs. The datasets in experiments are relatively small, it would be better to evaluate on larger KGs, such as YAGO3-10 and obgl-wikiKG2.

2. The hidden size of CFDenoiser is too large (such as 500，1600，2000). How about the dimension of embedding vectors? The efficiency issue of FDM is concerning.

3. Although the authors present a case study on FB15k-237, the reason for the significant performance gain on this dataset is still unclear. In fact, except FB15k-237, the superiority of FDM on the other 3 datasets is not obvious enough. The advantages of the diffusion perspective should be analyzed in depth. In addition, the effect of explicit conditional constraints is not fully discussed in the experiments.

**Questions:**

please see the weaknesses in the above review

**Reviewer Confidence:**

3: The reviewer is confident but not certain that the evaluation is correct

**Scope:**

3: The work is somewhat relevant to the Web and to the track, and is of narrow interest to a sub-community

---

### Official Review · Reviewer_gCsk · 2023-11-22

**Novelty:** 5
**Technical Quality:** 4

**Review:**

Summary:

This paper introduces a framework called Fact Embedding through Diffusion Model (FDM). FDM learns the distribution of plausible facts directly from the known knowledge graph, transforming the entity prediction task into a conditional fact generation task. It uses a Conditional Fact Denoiser to create fact embeddings from noised data. Experiments show that FDM achieves SOTA on FB15k-237 dataset.

Strengths:

1. State-of-the-art performance on FB15k-237 dataset.
2. Model architecture is quite novel.

Weakness:
1. The motivation could be clarified further. The issue of explicit scoring functions being inadequate for modeling KG patterns needs more comprehensive discussion.
2. It would be beneficial if the authors included data on training and inference times, along with the size of the model parameters for the proposed FDM, comparing these aspects with baseline methods.
3. The ablation study appears overly simplistic, with merely substituting a condition encoder for a transformer-based architecture. More extensive ablation studies are suggested to substantiate the optimal model design.
4. In the case study, using only a single sample does not sufficiently demonstrate FDM's effectiveness. Additional analysis, particularly on the advantages of the diffusion architecture, would be valuable.
5. The current experiments are generic and could apply to any KGE work. Conducting experiments specifically tailored to the diffusion model design would strengthen the paper's contribution.

**Questions:**

N/A

**Reviewer Confidence:**

3: The reviewer is confident but not certain that the evaluation is correct

**Scope:**

3: The work is somewhat relevant to the Web and to the track, and is of narrow interest to a sub-community

---

### Official Review · Reviewer_iToA · 2023-11-23

**Novelty:** 5
**Technical Quality:** 5

**Review:**

The authors propose a graph embedding method for knowledge graph completion called Fact Embedding through Diffusion Model (FDM). The method takes as inspiration the diffusion models approach used e.g., for image generation models, and applies its principles to the graph embeddings training task. The model consists of two processes: the forward process consequently applying gaussian noise to the initial input and the reverse denoising diffusion process to generate the target fact embedding from noised data. The authors use an MLP-based denoiser as opposed to a transformer model. The authors test the method on 4 benchmark datasets (FB15k-237, WN18RR, Kinship, and UMLS) and show performance improvements in comparison with the state of the art.

The topic and the method are clearly relevant for the conference and the track. Adapting the diffusion models’ principles for graph representation learning is an interesting idea. The method looks promising, particularly, the comparative evaluation results.
There are, however, some aspects which make it difficult to evaluate properly the added value of the approach. First, applying diffusion models’ principles to graph data is not a completely novel idea [1] and the related work section should mention existing work in this area and highlight the difference of the proposed approach to existing algorithms.

There are also well-known issues with commonly used benchmarks, in particular, inherent biases of FB15k-237 and WN18RR [2, 3], such as symmetric relations, over-representation of “popular” entities becoming default answers, etc. Given the popularity of the known benchmarks it is hard to avoid using them for comparison tests, but at least it would be interesting to see a more in-depth discussion, e.g., on what kind of triples the proposed method tends to outperform the state of the art and what kind of triples are handled less well (e.g, the observation that FDM performs worse on WN18RR, which was found to be less affected by the 3 types of biases selected in [2])

One question regarding the evaluation: why the comparative evaluation included different sets of methods in Tables 2 and 3 (e.g., TransE only in Table 2, but ComplEx only in Table 3)?

Typos:

-	P.3 $X^\tau = [𝑿^h;𝑿^r ;𝑿^t ] \in R^{2×𝑟+𝑒}$ $\rightarrow$ shouldn’t it be $2×e+r$ ?
-	Fig. 1 (a) FDM Architecther -> FDM Architecture

1.	Zhang, M. et al. A Survey on Graph Diffusion Models: Generative AI in Science for Molecule, Protein and Material. https://arxiv.org/abs/2304.01565
2.	Rossi, A. et al. Knowledge Graph Embeddings or Bias Graph Embeddings? A Study of Bias in Link Prediction Models. 2022
3.	Akrami, F. et al. Realistic Re-evaluation of Knowledge Graph Completion Methods: An Experimental Study. 2020

**Questions:**

(see the review section)
- How is the proposed method different from other algorithms applying diffusion models' principles to knowledge graph data?
- Given the inherent biases of common benchmarks, is it possible to evaluate how affected is the proposed method by them?
- Minor: why the comparative evaluation includes different sets of methods in Tables 2 and 3?

**Reviewer Confidence:**

2: The reviewer is willing to defend the evaluation, but it is likely that the reviewer did not understand parts of the paper

**Scope:**

4: The work is relevant to the Web and to the track, and is of broad interest to the community

---

### Official Review · Reviewer_u5qZ · 2023-11-28

**Novelty:** 5
**Technical Quality:** 5

**Review:**

The paper presents a link prediction method based on fact embeddings with diffusion models. The model uses Fact Embedding through Diffusion Model with Denoising Diffusion Probabilistic Models to learn the distribution of plausible facts. It casts the entity prediction task into a conditional fact generation task. The method is evaluated on 4 standard benchmark datasets against a large variety of recent link prediction models. The presented model outperforms all of them by a rather large margin.

Overall, this is a novel and highly technical paper, with a state-of-the-art result on a variety of benchmark datasets. The extensive experimental results show the advantages of the presented methods and explain different design choices by additional ablation studies.



**Strengths:**

-	Well-motivated paper
-	State-of-the-art results on a variety of common benchmark datasets
-	Extensive related work
-	Informative ablation studies for some of the design choices in the model


**Weaknesses:**

-	Results only on smaller benchmark datasets

**Questions:**

-	Why did you not use any of the larger datasets, e.g. Codex, Wikidata5m, and YAGO3-10?
-	Do you have ideas for future work?
-	What are the shortcomings of your method?

**Reviewer Confidence:**

2: The reviewer is willing to defend the evaluation, but it is likely that the reviewer did not understand parts of the paper

**Scope:**

4: The work is relevant to the Web and to the track, and is of broad interest to the community

---

### Decision · Program_Chairs · 2024-01-22

**Decision:**

Accept

**Comment:**

The reviewers have noted the following strengths associated with the paper:

 * Good motivation / relevant problem (though one reviewer believes the motivation could be improved).
 * Promising experimental results.
 * Good experimental design (though one reviewer finds the ablation study simplistic).
 * Extensive related work.
 * Good relevance to conference and track.
 * Novel approach.
 * The paper is relatively easy to follow.

 And the following weaknesses:

 * Only small datasets are considered when larger datasets are available.
 * Lacking detail about related works involving diffusion models.
 * Missing qualitative analysis (considering limitations of benchmarks used).
 * Missing training and inferencing times.
 * Missing experiments tailored to the diffusion model design.
 * Concerns about efficiency.

 In their responses, the authors address some of the aforementioned concerns, leading some reviewers to improve their score.

 Overall, the scores and comments of the reviewers consistently lean towards a positive evaluation (while not being overly positive) concerning relevance, technical quality and novely. Given the consistently positive reviews, and the lack of any clear reason against, I recommend an Accept.